# Laying Hens Biochar Diet Supplementation—Effect on Performance, Excreta N Content, NH₃ and VOCs Emissions, Egg Traits and Egg Consumers Acceptance

**Kajetan Kalus** [1,*] , **Damian Konkol** [1] , **Mariusz Korczyński** [1] , **Jacek A. Koziel** [2] and **Sebastian Opaliński** [1]

1   Department of Environment Hygiene and Animal Welfare, Wroclaw University of Environmental and Life Sciences, 51-630 Wrocław, Poland; damian.konkol@upwr.edu.pl (D.K.); mariusz.korczynski@upwr.edu.pl (M.K.); sebastian.opalinski@upwr.edu.pl (S.O.)

2   Department of Agricultural and Biosystems Engineering, Iowa State University, Ames, IA 50011, USA; koziel@iastate.edu

*   Correspondence: kajetan.kalus@upwr.edu.pl

**Abstract:** Sustainable solutions for intensive poultry production can help farmers, rural communities, consumers, and regulatory agencies. This study assessed supplementation of laying hens diet with beechwood biochar (BC, 1~2%) and BC–aluminosilicates–glycerin mixture (BCM, 1.5~3%) to lower the environmental impact while maintaining egg quality. The effect on feed intake, laying performance, egg quality, the sensory quality of hardboiled eggs, ammonia (NH₃) and volatile organic compound (VOC) emissions from excreta, and the excreta composition, were evaluated. A total of 90 hens were distributed into 30 cages and divided into five groups (*n*=6 replications). BC addition increased daily feed intake, while 1.5% BCM addition reduced it. The influence on egg parameters was positive, with a 6% increase in laying performance, up to 10% and 6% increase in shell resistance to crushing and shell thickness, respectively. The sensory analysis demonstrated no significant differences between all treatments. Excreta total N content was numerically lower due to the treatments (by 4~20%); its pH increased (not significantly), while no effect on ammoniacal N and dry matter content was observed. Most of the investigated treatments had a numerically positive (not statistically significant) effect on NH₃ reduction. The reduction of VOC emissions was ambiguous and not statistically significant.

**Keywords:** biocoal; poultry; feed; additive; volatile organic compounds; air quality; waste management; manure

## 1. Introduction

Poland has the third-largest number of laying hens in the European Union, with over 48.5 million of the birds in 2018 and 649 thousand tonnes of eggs (sixth place) produced in 2019 [1]. Such intensive poultry production carries with it an environmental impact associated with emissions of ammonia (NH₃) and volatile organic compounds (VOCs). Those gaseous emissions are of concern to workers, local communities, birds and the environment. Additionally, some VOCs have an unpleasant odor that can be of a nuisance, impacting public relations [2]. Mitigation of those emissions and improvement of air quality inside poultry houses without a negative impact on production are severe challenges for the industry.

There are many approaches to mitigate gaseous emissions from livestock production. However, most of them have been evaluated for swine production [3–6]. Those methods include dietary manipulation, manure additives, or air cleaning and manure treatment systems [7–14]. An additional

challenge is that only ~25% of technologies were farm-tested, i.e., the majority of them are not ready for commercial-scale applications [4].

Biochar as a feed additive for farm animals (goats, dairy cows, broilers and layers) has been known for some time. The results of such dietary manipulation among different species are, in general, positive, i.e., in terms of better feed conversion ratio, digestion, weight gain or mitigation of greenhouse gas (GHG) emissions from manure [15,16]. However, the use of biochar as a feed additive in poultry is still a relatively new research subject with very promising application potential.

"Biochar" is a term for substances with quite a wide range of physicochemical properties, e.g., pH, morphology or the residual content of chemicals from feedstock and its thermal decomposition, with the main purposes of utilization other than as an energy source. Biochar is obtained via torrefaction or pyrolysis of an organic material where biomass is decomposed by heat in an anoxic environment. The result is gaseous products, (sometimes) pyrolytic oil and carbon-rich solid remainder termed biochar, char, or biocoal. Biochar indicates that the feedstock was biomass, sometimes a byproduct of syngas or biorenewable fuel production [15]. Thus, opportunities exist to valorize biochar to continue fuel interest in the waste-to-carbon, waste-to-energy, circular economy and sustainable agriculture research.

Biochar properties are determined by their composition that is strongly dependent on its feedstock material [17]. Even simple experiments visualizing the behavior of two types of biochar: highly porous and alkaline or acidic biochar, when applied to water, reveal spatial and temporal differences in the pH at the air–water–biochar interface [18]. Carbon as a dietary supplement can bring many benefits during biotransformation processes in the gastrointestinal tract [19]. However, differences in biochar properties can lead to different effects on different animal species. For example, biochar diet inclusion could enhance laying hens production parameters and improve (decrease) feed conversion ratio [20,21], while for broiler chickens, the results were opposite [22,23], resulting in a decrease in body weight gain and increase in feed conversion ratio.

In this experiment, evaluation of the influence of biochar inclusion on laying hens diet was conducted. Beechwood biochar (BC) was used as a marketed animal feed additive. The biochar was obtained by the slow pyrolysis (maximum T = 550 °C) and activated with gases produced during the process. BC–aluminosilicates–glycerin mixture (BMC) named "NowiCarbon" was also tested as an example of marketed biochar-based feed additive, both manufactured by Ekomotor Ltd. (Wrocław, Poland). The BCM is claimed by the manufacturer to reduce toxins and mycotoxins in the feed, reduce herd mortality, improve feed conversion ratio, and enhance herd's health and productivity with the recommended dose of 0.3%. In the study, birds' performance and excreta parameters were evaluated, such as daily feed intake, laying performance and egg quality parameters (average mass, shell thickness, resistance to crushing and yolk color), ammonia and volatile organic compound (VOC) emissions from excreta and excreta composition. In addition, sensory analysis of hardboiled eggs was conducted for their appearance, smell, white and yolk texture, yolk color and egg taste, for the first time for a biochar dietary inclusion.

## 2. Materials and Methods

### 2.1. Ethical Approval

Polish law, particularly an Act of 15.01.2015 on the Protection of Animals Used for Scientific and Educational Purposes, specifies terms and conditions on the protection of animals used for scientific or educational purposes, including conditions when an Ethical Approval is required. Ethical Approval is not required for veterinary services within the scope of the Act of 18.12.2003 on animal treatment facilities, as well as agricultural activities, including rearing or breeding of animals, carried out in accordance with the provisions on the protection of animals; and activities that, in compliance with the veterinary medicine practice, do not cause pain, suffering, distress or permanent damage to the body of animals, to an extent equal to a needle stick, or more intense. In this research:

- the animals were maintained in the standard production conditions,
- the animals were not exposed to pain and suffering in any way,
- no blood samples were taken,
- sampling of the excreta was not harmful to birds in any way.

Thus, the experiment did not require Ethical Approval under the abovementioned applicable law.

## 2.2. Laying Hens and Feed Additives

A total of 90 laying hens (Lohmann Brown), 20 weeks of age, were randomly distributed into 30 cages and divided into five groups (six replicates). Three hens per cage ($0.125\ m^2 \times hen^{-1}$) were housed for 13 weeks at the Agricultural Experimental Plant "Swojec" of Wrocław University of Environmental and Life Sciences, within a standard 3-tier furnished cages system, with all three tiers (top, middle and bottom) consisting of two rows. Feed was supplied daily for each cage and the average daily feed intake (ADFI) was estimated once a week. Evaluation of the additives' effectiveness on ADFI was made on the basis of the relative increase value (RIV, %) calculated as the ratio of the difference between the control and treatment parameter means.

Two additives were used in the experiment: beechwood biochar (BC) and the mixture consisting of 67% of the same BC, 24% of aluminosilicates as an anticaking agent and 9% of glycerin as an antidusting agent (BCM). Properties of the additives are presented in Table 1. Treatment feed mixtures were prepared weekly in 5 L plastic jars, separately for each cage to ensure even distribution of the investigated additives. All laying hens were fed with the same basal diet formulated according to nutrient recommendations for laying hens (Tables 2 and 3). The control group (C) was fed only with the basal diet. Treatment groups were fed with the addition of BC in the amount of 1% and 2% by mass (groups BC1 and BC2, respectively) and with the addition of BCM in the amount of 1.5% and 3% by mass (groups BCM1 and BCM2, respectively). The mass of BC in the BCM1 and BCM2 groups was numerically the same as in the corresponding BC1 and BC2 groups.

**Table 1.** Properties of additives used in the experiment.

| Property | Additive | |
|---|---|---|
| | **Beech Wood Biochar** | **Aluminosilicates** |
| pH | 9.0 | 8.7 |
| Fiber Content | 75% | – |
| Iodine Index | $140\ mgI_2 \times g^{-1}$ | – |
| Methylene Index | 3 ml | – |
| Bulk Density | $350.0\ g \times dm^{-3}$ | $741.2\ g \times dm^{-3}$ |
| Ash Content | no more than 10.0% (by mass) | – |
| Moisture Content | no more than 8.0% (by mass) | no more than 6.2% (by mass) |
| Volatile Matter | 14% | – |
| Granulation | 0.0–1.0 mm | – |
| Specific Surface Area | $140\ m^2 \times g^{-1}$ | – |
| Mineral Composition | – | Clinoptilolite—up to 83% (by mass) Feldspars—up top 5% (by mass) |
| 0.600 mm Sieve Screenings | – | 0.2 |
| <0.125 mm Sieve Screenings | – | 7.1 |

**Table 2.** Nutrient content of the basal diet.

| Nutrient Content | |
|---|---|
| Metabolizable energy | 11.3 MJ $\times$ kg$^{-1}$ |
| Crude protein | 16.52% |
| Crude fiber | 4.95% |
| Crude ash | 7.34% |
| Crude fat | 2.39% |
| Lysine | 0.73% |
| Methionine | 0.33% |
| Calcium | 3.75% |
| Phosphorus | 0.50% |
| Sodium | 0.14% |

**Table 3.** Ingredients and the chemical composition of the basal diet.

| Item | Amount (%) |
|---|---|
| Corn | 24.94 |
| Wheat | 29.80 |
| Postextraction soybean meal | 9.10 |
| Sunflower seed meal | 8.60 |
| Limestone | 9.23 |
| Corn dried distillers grains with solubles | 5.00 |
| Wheat-mix [1] | 3.00 |
| Guar meal | 3.00 |
| Oat | 3.00 |
| Fat | 2.60 |
| Monocalcium phosphate | 0.39 |
| L-Lysine sulfate | 0.24 |
| Acidifier [2] | 0.25 |
| Sodium bicarbonate | 0.20 |
| Mineral–vitamin premix [3] | 0.23 |
| Noniodized sodium chloride | 0.18 |
| DL-Methionine | 0.10 |
| Microsorbent | 0.06 |
| Choline chloride | 0.04 |
| L-Threonine | 0.02 |
| NSP enzymes | 0.01 |
| Phytase | 0.01 |

[1] A mixture of separated wheat germ, wheat endosperm and wheat skin. [2] Formic acid, propionic acid, ammonium formate, ammonium propionate. [3] Provides (mg $\times$ kg$^{-1}$ of diet): Co, 1 (as CoCO$_3$); Cu, 9 (as CuSO$_4\cdot$5H$_2$O); Fe, 30 (as FeSO$_4\cdot$H$_2$O); Mn, 80 (as MnO$_2$); Se,0.4(as SeO$_3\cdot$5H$_2$O); Zn, 80 (as ZnO); butylated hydroxytoluene, 0.6; butylated hydroxyanisole, 0.06; ethoxyquin, 0.1. NSP = nonstarch polysaccharides.

During the first four weeks of the experiment, no parameters were measured so that the hens could adapt to the new surroundings and the treatments could take effect.

### 2.3. Egg Quality

Eggs were collected and counted daily (86 eggs on average for the whole experiment) and weighed twice a week.

At the end of the experiment, sensory analysis of the eggs was conducted by 75 panelists (a total of 375 eggs, 5 eggs per panelist with 1 egg per group) randomly selected and trained from among the university employees and students, to evaluate the influence of used additives on the eggs' overall appearance, i.e., taste, smell, white and yolk texture and color. Sensory analysis was conducted under a completely randomized design, where the panelists were unaware of the egg samples' origin and hardboiled eggs samples were rated on a five-point scale from 1 (the worst) to 5 (the best) for each parameter.

A total of 435 eggs (87 per group) were analyzed for resistance to crushing, shell thickness and yolk color. Resistance to crushing was measured by Egg Force Reader[TM] (Orka Food Technology Ltd., West Bountiful, UT, USA), shell thickness was measured with a micrometer screw gauge and yolk color was evaluated with a La Roche scale yolk color fan.

Evaluation of the additives' effectiveness on ADFI and egg parameters was made on the basis of the relative increase value (RIV, %) calculated as the ratio of the difference between the control and treatment parameters' means.

### 2.4. Excreta Sampling

Samples of the hens' excreta were collected, at even intervals, twice a week from the belts under the cages and the belts were cleaned right after the collection. The samples were taken in the following manner:

- during the first week, 3 C + 3 BC1 and 3 C + 3 BCM1 samples were collected for $NH_3$ analysis;
- during the second week, 3 C + 3 BC2 and 3 C + 3 BCM2 samples were collected for $NH_3$ analysis;
- during the third week, 2 C + 2 BC1 + 2 BC2 and 2 C + 2 BCM1 + 2 BCM2 samples were collected for VOCs analysis;
- the above sequence was repeated 3 times. Therefore the experiment lasted for 9 weeks, resulting in a total of 9 comparative replications for $NH_3$ analysis and 3 replications of VOCs analysis.

During the first two weeks of the $NH_3$ sampling, 100 g of the excreta from under 12 cages (6C+6 Treatment) were collected for $NH_3$ emission analysis. The excreta from three separate tiers (top, middle and bottom) were thoroughly mixed (composited) with the use of a mechanical stirrer, resulting in three 200 g replications of C excreta and three 200 g replications of particular treatment excreta during each sampling. For $NH_3$ analysis, with four treatment groups and excreta sampling twice a week, one sampling trial took two weeks, resulting in 3 replications per each treatment, on one trial.

In the third week of VOCs sampling, 50 g of excreta from under 18 cages (6C+6Treatment1+6Treatment2) were collected for VOCs analysis. Excreta from each group (6 cages) was mixed (composited) thoroughly with the use of a mechanical stirrer, resulting in 300 g of control excreta, 300 g of the treatment excreta and another 300 g of the same type of treatment excreta with the different concentration of the additive used. With two types of additives and excreta sampling twice a week, one trial of VOCs analysis took one week.

Graphical visualization of the experimental design concerning excreta sampling is presented in Figure 1.

At the end of the experiment, 100 g of the excreta from under each cage (a total of 30 samples) was collected to compare total N and ammoniacal N (N-$NH_4^+$) content, pH and dry mass content between all the treatment groups and control. The excreta analyses were carried out by the accredited Chemical and Agricultural Research Laboratory in Wrocław, Poland.

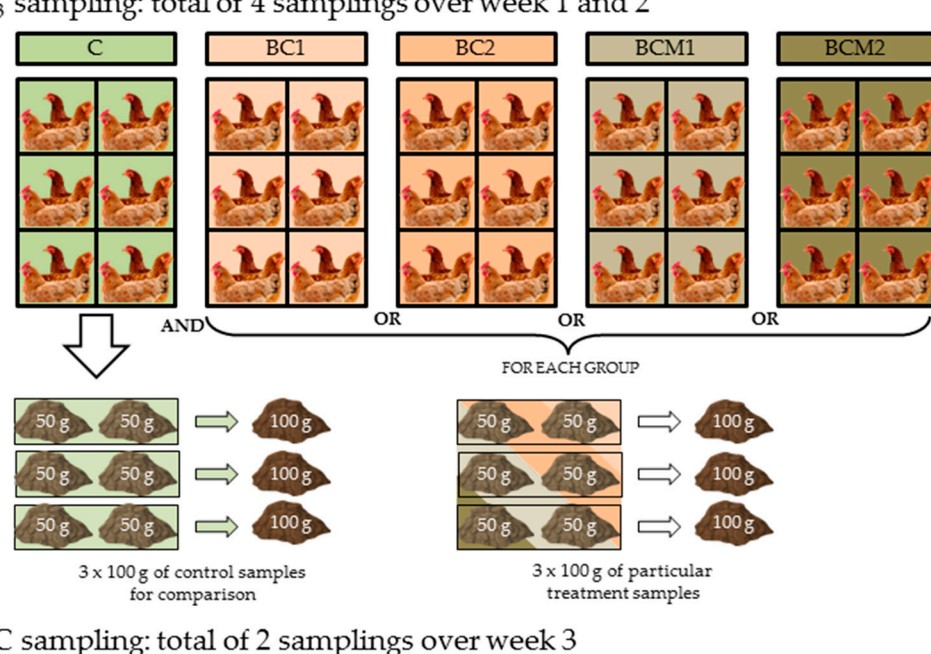

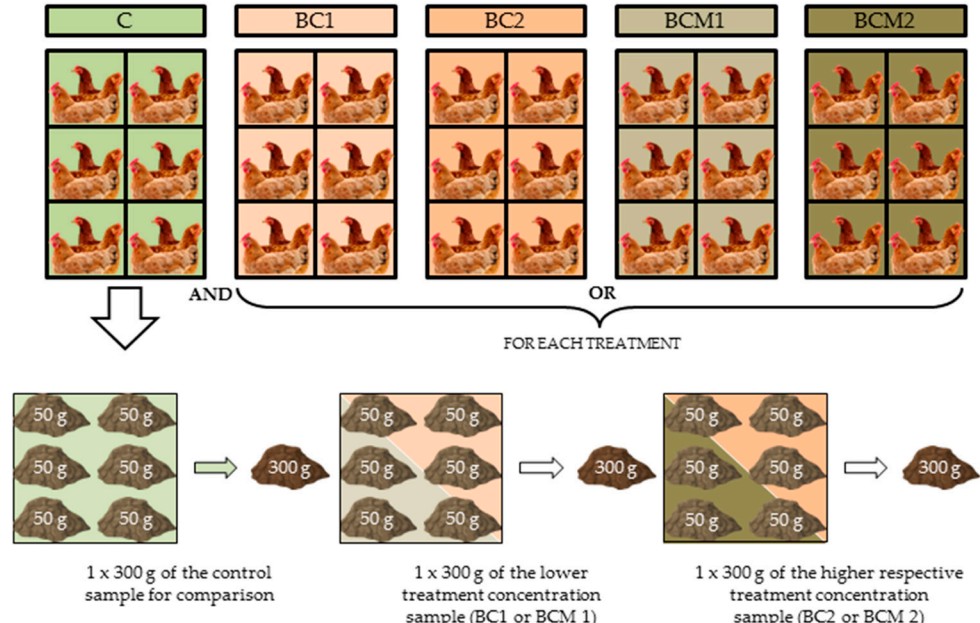

**Figure 1.** Visualization of the excreta sampling process.

*2.5. Gas Sampling and Chemical Analysis*

2.5.1. Ammonia

After the collection, samples were taken directly into the laboratory, put into glass treatment containers and the containers were closed for 30 min to stabilize the conditions inside. Afterward, $NH_3$ was sampled by portable aspirators. The excreta headspace air was pulled through impingers containing 0.1M $H_2SO_4$ for 10 min at $30 \text{ L} \times \text{min}^{-1}$. $NH_3$ concentration in the excreta headspace was measured right after sampling and 24 h later. The content of $NH_3$ was determined using Nessler's reagent and a UV-3100 PC spectrophotometer (VWR International, Leuven, Belgium) according to the Polish Standard Method (standard no. PN-71/Z-04041 "Determination of ammonia content in the

air"). Evaluation of effectiveness in $NH_3$ emission mitigation was made on the basis of the relative reduction value (RRV, %) calculated as the ratio of the difference between the control and treatments mean (of 9 replicates) $NH_3$ concentrations.

### 2.5.2. Volatile Organic Compounds

Excreta samples were taken directly into the laboratory after collection, placed into 1000 mL beakers, covered with Parafilm and heated (40 °C) for 20 min in a laboratory drying oven in order to stabilize the conditions inside and improve the sensitivity of the measurements. Afterward, manual solid-phase microextraction (SPME) with Stable-Flex 50/30 μm divinylbenzene/Carboxen/polydimethyl-siloxane (DVB/CAR/PDMS) fiber (Supelco, Bellefonte, PA, USA) was used for extraction of odorous compounds from the headspace inside the beakers. An optimal SPME sampling time of 15 min was determined during preliminary experiments. Collected gas samples were analyzed using a gas chromatograph–mass spectrometer (GC–MS) (Agilent Technologies, Santa Clara, CA, USA). VOCs were tentatively identified on the basis of comparative analysis between determined mass spectra and the National Institute of Standards and Technology (NIST02) MS library. The GC–MS system components and basic GC oven temperature program are described in Table 4.

**Table 4.** Gas chromatography–mass spectrometry parameters for volatile organic compounds separation and analysis.

| Oven Temperature Program | | |
|---|---|---|
| | | Time |
| Initial temp. | 40 °C | 0.00–2.00 min |
| Temp. ramp | $4\,°C \times min^{-1}$ | 2.00–52.00 min |
| Final temp. | 240 °C | 52.00–62.00 min |
| Post-run temp. | 40 °C | – |
| **Front Inlet Parameters** | | |
| Mode: | Splitless | |
| Initial temp: | 240 °C | |
| **Gas Chromatography Capillary Column Parameters** | | |
| Model Number: | Agilent 112-88A7 HP-88 | |
| Nominal length: | 100.0 m | |
| Nominal diameter: | 250.00 μm | |
| Nominal film thickness: | 0.20 μm | |
| Mode: | Constant flow | |
| Initial flow: | $1.5\,mL \times min^{-1}$ | |

Evaluation of air purification effectiveness was made on the basis of the RRV determined for each tentatively identified odorous VOC. The RRV was calculated as the ratio of the difference between the control and treatments mean (of 3 replicates) peak height counts of the tentatively identified odorous VOCs [12]. Peak heights were used for comparison instead of peak areas, to improve the integration of asymmetric or low chromatographic peaks.

### 2.6. Statistical Analysis

Statistical analysis of the results was carried out using Statistica 13.1 software (TIBCO Software Inc., Palo Alto, CA, USA). Data were tested for normality with the Shapiro–Wilk test. If the distribution was normal, a one-way analysis of variance was performed with the differences between the groups assessed using the Tukey test. If the distribution was not normal, the Kruskal–Wallis test was carried out. Differences were considered significant at $p$-value < 0.05.

## 3. Results

### 3.1. Ammonia

Mean $NH_3$ concentrations, relative reduction value (RRV), standard error of the mean (SEM) and *p*-value for all of the treatment groups evaluated ~1 h and 24 h after sampling, are presented in Table 5. Compared with the control group, $NH_3$ concentration in the treated excreta headspace, measured ~1 h after sampling, was 81% higher for the BC1 group, 24% lower for the BC2 group, no differences for the BCM1 group and 66% lower for the BCM2 group. Only the difference between BC1 and control groups was statistically significant ($p < 0.05$). Twenty-four hours after sampling, $NH_3$ concentrations were numerically 17% and 18% lower, 1% higher and 5% lower, for the BC1, BC2, BCM1 and BCM2 groups, respectively, but the differences were not statistically significant ($p > 0.05$). The statistical insignificance, together with high standard deviations, could result from the fact that gaseous emissions from excreta are associated with microbial activity in the excreta, which is difficult to control, predict and maintain between all replications. Nonetheless, the results show some preliminary trends of biochar–excreta interactions. A higher number of replications are recommended when working with manure-like matrixes.

**Table 5.** Mean ammonia concentrations in treated excreta headspace.

| $NH_3$ Concentrations (mg·m$^{-3}$) ($n = 9$) ~1 h After Sampling | | | | |
|:---:|:---:|:---:|:---:|:---:|
| **Mean** | | **RRV (%)** | **SEM** | ***p*-Value** |
| **C** <br> 23.66 [a] | **BC1** <br> 42.76 [b] | −81 | 2.75 | <0.00100 |
| **C** <br> 40.22 | **BC2** <br> 30.61 | 24 | 5.67 | 0.1850 |
| **C** <br> 21.05 | **BCM1** <br> 21.15 | <0.10 | 2.18 | 1.000 |
| **C** <br> 73.80 | **BCM2** <br> 27.18 | 66 | 14.6 | 0.07730 |
| **24 h After Sampling** | | | | |
| **Mean** | | **RRV (%)** | **SEM** | ***p*-Value** |
| **C** <br> 198.8 | **BC1** <br> 164.0 | 17 | 26.1 | 0.0773 |
| **C** <br> 114.9 | **BC2** <br> 94.59 | 18 | 8.46 | 0.0520 |
| **C** <br> 150.7 | BCM1 <br> 152.4 | −1.0 | 33.1 | 0.929 |
| **C** <br> 257.5 | **BCM2** <br> 244.9 | 5.0 | 14.2 | 0.671 |

**Notes:** C = control group; BC1 = 1% by mass of beech wood biochar; BC2 = 2% by mass of beech wood biochar; BCM1 = 1.5% by mass of beech wood biochar-based additive; BCM2 = 2% by mass of beech wood biochar-based additive; RRV = relative reduction value; SEM = standard error of the mean. The mass of beech wood biochar in the BCM1 and BCM2 groups was the same as in the corresponding BC1 and BC2 groups. [a,b] Mean values within the same row with no common superscript indicate significant differences ($p < 0.05$). Mean values with no superscript are not significantly different from any other values. Negative values indicate an increase in the $NH_3$ concentration compared to the control group.

### 3.2. Volatile Organic Compounds

A total of nine VOCs were identified in the analyzed gas samples emitted from the treated/untreated poultry excreta. The compounds with their matching percentage of identity assignment in the MS spectral database, GC column retention times ($R_T$) and mean (of three replicates) peak height (H), along with standard deviations and relative reduction values (RRV) compared to the control group (C), are shown in Table 6. The results of gas chromatography analysis are ambiguous, i.e., no statistically significant differences between any of the control/treatment groups were determined due to the high standard deviations of the results. Comprehensive and detailed chromatographic VOC

analyses are costly, time-consuming and require sophisticated sampling, sample preparation and analysis equipment, all of which were limitations of this study, as the methodology was designed to investigate many different aspects of the treatments used. Nonetheless, it is recommended that when working with manure-like matrixes, a higher number of replications should be considered. The results show an apparent numerical trend that lower doses of the additives used have better VOC mitigation potential, with the BC1 treatment reductions from 9% for 3-octanone up to 96% for (E,E,)-2,4-heptadienal, while BCM1 treatment showed reductions from 1% for 2-propenoic acid and 3-(4-methoxyphenyl)-2-ethylhexyl ester up to 91% for disulfide dimethyl. BC1 treatment showed a slightly better overall deodorizing potential of 40%, expressed as an average reduction calculated as a mean reduction of every single compound, while BCM1 treatment overall deodorizing potential was 37%. BC2 and BCM2 showed worse overall deodorizing potential, with 31% and 16% respectively, where emission reductions ranged from −9% for 2-butanone up to 82% for (E,E,)-2,4-heptadienal for BC2 treatment and from −63% for 3-octanone up to 97% for dimethyl disulfide.

### 3.3. Average Daily Feed Intake and Egg Parameters

All of the results, presented in Table 7, consist of a comparison of treatment groups with the control group. (1) ADFI of the biochar-fed laying hens for the BC1, BC2, BCM1 and BCM2 groups was 4% and 6% higher, 3% lower and 3% higher, respectively. The differences were statistically significant between control and BC2, BCM1 groups. Also, differences were significant between BCM1 and BC1, BC2 groups. The significantly lower ADFI in the BCM1 group may be an effect of glycerin present in the additive, being an additional energy source for the birds, vice versa for the biochar-only additives. (2) Laying performance had improved for all of the treatment groups by 6%, with all the differences being statistically significant. (3) Average egg mass improved for BC1, BC2, BCM1 and BCM2 groups by 2%, 3%, 4% and 4%, respectively, with all the differences also being significant and additionally between BC1 and BC2 groups. (4) Shell resistance to crushing had improved for BC1, BC2, BCM1 and BCM2 groups by 10%, 6%, 9% and 9%, respectively, but only the difference between control and BC1 groups was significant. (5) Shell thickness had improved for all BC1, BC2, BCM1 and BCM2 groups by 4%, 1%, 6% and 4%, respectively, with all the differences being statistically significant, also additionally between BC2 and BCM1 groups. (6) Yolk color grade for the BC1, BC2, NC1 and NC2 groups was 5% lower, 2% higher, 2% and 4% lower, respectively.

Panelists' mean grades for egg appearance, smell, white texture, yolk texture, yolk color and taste for all of the analyzed parameters, along with the SEM and *p*-values, are presented in Table 8. The mean grades ranged from 3.21 to 3.86. Sensory analysis of the hardboiled eggs demonstrated that there are no significant differences between all of the treated/untreated eggs whatsoever, except only for the difference in yolk color between BCM1 and BCM2 groups.

### 3.4. Excreta Properties

Mean values for all of the analyzed excreta parameters along with the SEM and *p*-values are presented in Table 9. The mean pH values of the excreta analyzed at the end of the experiment ranged from 6.86 for the control group up to 7.35 for the BC1 group. Although the differences are not significant, it is worth noting that the pH of excreta from all of the treatment groups was higher than control pH, most likely due to the use of the biochar and aluminosilicates with alkaline pH (9.0 and 8.7, respectively). Total N content ranged from 1.28% for the BC1 group to 1.60% for the control group, with the statistical differences between control and BC1, BCM2 groups. The reduction of total N content resulted likely due to the loss of nitrogen from the transformation of ammonium ions ($NH_4^+$) into gaseous $NH_3$ that is associated with pH increase [24], which is consistent with higher emission of $NH_3$ from BC1 excreta. Ammoniacal nitrogen ($N-NH_4^+$) content ranged from 0.34% for the BC1 group to 0.44% for the BC2 group, with significant differences only between BC1 and BC2 groups. Dry matter content ranged from 28.89% for the BC1 group to 29.18% for the BCM1 group, with no significant differences between all of the groups.

**Table 6.** Effect of investigated treatments on tentatively identified volatile organic compounds (mean, *n* = 3).

| | Compound | $R_T$ (min) | Spectral Match with MS Database | BC | | | | | | | | BCM | | | | | | | | |
|---|---|---|---|---|---|---|---|---|---|---|---|---|---|---|---|---|---|---|---|---|
| | | | | C | | BC1 | | | BC2 | | | C | | BCM1 | | | BCM2 | | |
| | | | | H | SD | H | SD | RRV | H | SD | RRV | H | SD | H | SD | RRV | H | SD | RRV |
| 1. | dimethyl sulfide | 8.49 | 87% | 50,2785 | 395,459 | 324,629 | 147,682 | 35% | 369,560 | 273,998 | 26% | 573,939 | 399,258 | 433,339 | 156,240 | 24% | 577,347 | 172,643 | −1% |
| 2. | 2-butanone | 11.88 | 70% | 217,069 | 134,491 | 158,776 | 22,417 | 27% | 236,992 | 61,660 | −9% | 216,774 | 175,651 | 86,768 | 55,413 | 60% | 207,902 | 20,311 | 4% |
| 3. | dimethyl disulfide | 14.66 | 96% | 92,251 | 44,266 | 9446 | 1803 | 90% | 25,376 | 8709 | 72% | 598,706 | 703,584 | 53,718 | 32,983 | 91% | 20,626 | 6385 | 97% |
| 4. | 3-octanone | 22.11 | 74% | 57,025 | 44,743 | 51,715 | 20,416 | 9% | 61,629 | 2698 | −8% | 46,103 | 17,439 | 44,084 | 13,810 | 4% | 75,375 | 6154 | −63% |
| 5. | (E,E,)-2,4-heptadienal | 24.92 | 95% | 95,106 | 43,448 | 3516 | 640 | 96% | 17,350 | 11,078 | 82% | 38,644 | 13,879 | 24,432 | 9342 | 37% | 45,064 | 28,653 | −17% |
| 6. | 1-butene, 4-isothiocyanato ether | 27.61 | 78% | 10,925 | 3996 | 7293 | 664 | 33% | 5339 | 698 | 51% | 83,318 | 5276 | 27,462 | 3470 | 67% | 24,092 | 7726 | 71% |
| 7. | 2-propenoic acid, 3-(4-methoxyphenyl)-2-ethylhexyl ester | 39.81 | 98% | 80,264 | 11,322 | 55,922 | 17,909 | 30% | 52,674 | 14,801 | 34% | 56,739 | 7666 | 56,318 | 5789 | 1% | 58,466 | 13,493 | −3% |
| 8. | phenol | 40.18 | 94% | 109,856 | 5392 | 81,321 | 14,998 | 26% | 95,524 | 6849 | 13% | 132,732 | 60,649 | 85,576 | 9591 | 36% | 88,087 | 61,361 | 34% |
| 9. | 4-methyl phenol | 41.77 | 90% | 168,511 | 49,893 | 142,583 | 15,765 | 15% | 142,868 | 13,224 | 15% | 208,778 | 40,158 | 188,593 | 55,331 | 10% | 171,056 | 134,714 | 18% |

**Notes:** $R_T$, GC column retention time; C, control group; BC, biochar group; BCM, biochar–aluminosilicates–glycerin mixture additive; H, mean peak height; SD, standard deviation; RRV, relative reduction value—negative values indicate an increase in emission.

**Table 7.** Mean feed intake and egg parameters of biochar-fed laying hens.

| | | C | BC1 | BC2 | BCM1 | BCM2 | SEM | *p*-Value |
|---|---|---|---|---|---|---|---|---|
| | | **Egg Parameters (*n* = 87)** | | | | | | |
| Average daily feed intake (g) | Mean | 105.35 | 109.31 | 111.73 [a] | 102.53 [b] | 108.27 | 0.62 | <0.0010 |
| | RIV (%) | – | 4 | 6 | −3 | 3 | | |
| Laying performance (%) | Mean | 89.79 [b] | 95.59 [a] | 95.08 | 95.59 | 95.13 | 0.59 | 0.034 |
| | RIV (%) | – | 6 | 6 | 6 | 6 | | |
| Average egg mass (g) | Mean | 56.58 [b] | 57.63 [a] | 58.55 [a] | 59.07 [a,c] | 59.02 [b,c] | 0.090 | <0.0010 |
| | RIV (%) | – | 2 | 3 | 4 | 4 | | |
| Shell resistance to crushing (kg) | Mean | 5.02 [a] | 5.51 [b] | 5.34 | 5.49 | 5.45 | 0.050 | 0.036 |
| | RIV (%) | – | 10 | 6 | 9 | 9 | | |
| Shell thickness (mm) | Mean | 0.427 [b] | 0.446 | 0.433 [d] | 0.452 [a,c] | 0.444 [a,c,d] | 0.0010 | <0.0010 |
| | RIV (%) | – | 4 | 1 | 6 | 4 | | |
| Yolk color (LaRoche scale) | Mean | 13.22 [a,c] | 12.52 [d] | 13.46 [a] | 12.95 | 12.73 [b] | 0.050 | <0.0010 |
| | RIV (%) | – | −5 | 2 | −2 | −4 | | |

**Notes:** C = control group; BC1 = 1% by mass of beech wood biochar; BC2 = 2% by mass of beech wood biochar; BCM1 = 1.5% by mass of beech wood biochar-based additive; BCM2 = 2% by mass of beech wood biochar-based additive; RIV = relative increase value; SEM = standard error of the mean. The mass of beech wood biochar in the BCM1 and BCM2 groups was the same as in the corresponding BC1 and BC2 groups. RIV = relative increase value, calculated as the ratio of the difference between the control and treatment parameters' means. Negative values indicate a decrease of a particular parameter, compared to the control group. [a,b,c,d] Mean values within the same row with no common superscript indicate significant differences ($p < 0.05$). Mean values with no superscript are not significantly different from any other values.

**Table 8.** Sensory traits of hardboiled eggs after biochar supplementation.

| | C | BC1 | BC2 | BCM1 | BCM2 | SEM | *p*-Value |
|---|---|---|---|---|---|---|---|
| | **Eggs' Sensory Parameters (*n* = 75)** | | | | | | |
| Appearance | 3.59 | 3.48 | 3.57 | 3.82 | 3.39 | 0.05 | 0.104 |
| Smell | 3.53 | 3.48 | 3.68 | 3.67 | 3.52 | 0.04 | 0.694 |
| Egg white texture | 3.78 | 3.67 | 3.76 | 3.86 | 3.65 | 0.05 | 0.652 |
| Yolk texture | 3.61 | 3.40 | 3.55 | 3.67 | 3.21 | 0.05 | 0.0498 |
| Yolk color | 3.57 | 3.39 | 3.59 | 3.76 [a] | 3.21 [b] | 0.05 | 0.00380 |
| Taste | 3.59 | 3.40 | 3.67 | 3.71 | 3.48 | 0.06 | 0.358 |

**Notes:** C = control group; BC1 = 1% by mass of beech wood biochar; BC2 = 2% by mass of beech wood biochar; BCM1 = 1.5% by mass of beech wood biochar-based additive; BCM2 = 2% by mass of beech wood biochar-based additive; SEM = standard error of the mean. The mass of beech wood biochar in the BCM1 and BCM2 groups was the same as in the corresponding BC1 and BC2 groups. Each sensory trait is based on a 1–5 scale. [a,b] Mean values within the same row with no common superscript indicate significant differences ($p < 0.05$). Mean values with no superscript are not significantly different from any other values.

**Table 9.** Mean excreta parameters from biochar-fed laying hens.

| | C | BC1 | BC2 | BCM1 | BCM2 | SEM | *p*-Value |
|---|---|---|---|---|---|---|---|
| | **Excreta Parameters (*n* = 6)** | | | | | | |
| Total N (%) | 1.60 [a] | 1.28 [b] | 1.36 | 1.54 | 1.31 [b] | 0.04 | 0.00751 |
| Ammoniacal N (N-NH$_4^+$) (%) | 0.40 | 0.44 [a] | 0.34 [b] | 0.40 | 0.42 | 0.01 | 0.0355 |
| Dry matter (%) | 28.94 | 26.89 | 27.44 | 29.18 | 28.17 | 0.4 | 0.230 |
| pH | 6.86 | 7.35 | 6.91 | 6.90 | 7.15 | 0.07 | 0.0809 |

[a,b] Mean values within the same row with no common superscript indicate significant differences ($p < 0.01$). C = control group; BC1 = 1% by mass of beech wood biochar; BC2 = 2% by mass of beech wood biochar; BCM1 = 1.5% by mass of beech wood biochar-based additive; BCM2 = 2% by mass of beech wood biochar-based additive; SEM = standard error of the mean. Mean values with no superscript are not significantly different from any other values.

## 4. Discussion

### 4.1. Ammonia Emissions

BC1 addition to the laying hens feed resulted in a statistically significant increase in ammonia emission from excreta ~1h after sampling, while BC2 addition resulted in decreased $NH_3$ emission. At the same time, BCM1 showed no influence, but BCM2 resulted in a decrease of $NH_3$ emission. The situation changed ~24 h after excreta sampling, when BC1 and BC2 treatments showed a similar reduction of $NH_3$ emission, while BCM1 and BCM2 treatments had nearly no effect. Statistically significant increase in $NH_3$ emission from BC1 excreta ~1 h after sampling is consistent with the study by Prasai et al. [25] who reported $NH_3$ emissions from laying hens' excreta of 2% and 4% BC-amended feed was higher by 47% and 43%, respectively, over the course of the whole experiment (78 d), during which $NH_3$ release was rapid for the first three weeks. The results were supported by lower N levels in excreta of amended feed, which is also the case in this experiment as some of the treated excreta show lower levels of N content as well, likely due to the loss of nitrogen from a transformation of ammonium ions ($NH_4^+$) into gaseous $NH_3$ with increased pH. It is worth noting that BC introduced to the excreta as a feed amendment results in increased $NH_3$ emissions. At the same time, direct BC addition to the poultry litter actually helps to reduce those emissions [26]. To date, no other recent publications were found concerning the effect of dietary BC amendment on $NH_3$ emission from laying hens or even poultry manure.

### 4.2. VOC Emissions

The effect of BC amendment to the laying hens' diet was numerically positive and had mitigation potential, expressed as an average reduction calculated as a mean reduction of every single odorous compound, ranging from 16% to 40% for BCM2 and BC1, respectively. However, lower doses of the additives used show better VOC mitigation potential. The odorants identified in the study are in accordance with other research [8,9], where compounds such as dimethyl sulfide, 2-butanone, dimethyl disulfide, 3-octanone, phenol and 4-methyl phenol were also found in the poultry manure's headspace. To date, no recent publications were found concerning the effect of BC supplementation or other similar additives (i.e., aluminosilicates) on VOC emissions from laying hens or even poultry manure.

### 4.3. Bird Performance Parameters and Egg Parameters

ADFI of the investigated laying hens was significantly improved only in the BCM1 group, where it was higher for the other groups. However, laying performance and average egg mass were significantly improved for all of the treatment groups. Shell resistance to crushing was also numerically improved in all of the treatment groups, but only a difference in the BC1 group was significant. Yolk color grade was significantly lower for the BC1 and BCM2 groups, insignificantly lower for the BCM1 group and insignificantly higher for the BC2 group. Other research shows similar results. Experiments by [20] demonstrated that 4% BC diet supplementation insignificantly increased egg productivity by 1.2%, but significantly increased average egg weight by 3%, improved feed conversion ratio by 8% and lowered feed intake by 2%. Later experiments by [21] investigated 1%, 2% and 4% laying hens BC dietary addition. All of the treatments significantly improved feed conversion ratio by 9%, 14% and 12% and increased egg weight by 1%, 5% and 4%, respectively, with 2% and 4% BC addition having a statistically significant effect. Moreover, for hens of 36 weeks of age, 1%, 2% and 4% BC treatments increased eggshell weight by 6%, 13% and 13%, increased shell thickness by 4%, 6% and 11% and increased shell breaking strength by 10%, 11% and 19%, but also reduced yolk color score by 2%, 1% and 19%, respectively. Research by [27] showed that dietary supplementation with 1%, 2% and 4% of wood charcoal did not significantly affect any production parameters of laying hens during the whole experiment, except for the reduction of the number of cracked eggs by 24%, 38% and 65% for 1%, 2% and 4% of BC amendment, respectively, even though, on the other hand, the average weight of eggs was lower and shell weight was almost indifferent between the treatment groups and control.

*4.4. Excreta Properties*

All of the treatment groups showed numerically lower excreta N content with statistical differences between the control group and BC1, BCM2 groups. The content of ammoniacal nitrogen was not different between the treatment groups and control. Although there are no statistical differences, the pH of excreta from all of the treatment groups was numerically higher than the control pH. Dry matter content was numerically higher in the treatment groups compared to the control, with the BC1 and BCM2 having the highest value, but no statistically significant differences were observed between all of the groups. Again, a similar study on biochar supplementation of laying hens diet shows comparable results. In the study of [21],1%, 2% and 4% BC treatments reduced excreta N content by 9%, 17% and 26%, respectively, with 2% and 4% BC addition having a statistically significant effect. In the other experiment of [25], increasing contents of BC were associated with a decreased total N content of laying hens excreta, from 5.4% for 1%, by 5.0% for 2%, to 4.4% for 4% BC amendment.

*4.5. Sensory Analysis*

Sensory analysis of the hardboiled eggs demonstrated that there are no significant differences between all of the treated/untreated eggs whatsoever, which means there is likely no negative influence of the biochar dietary inclusion on the eggs. To date, no studies were identified that investigated consumers' acceptance of the hardboiled eggs after the amendment of laying hens diet with BC and this is the first such sensory analysis. However, a study by Ko et al. [28], by far the only one with a similar experiment, also showed that sensory traits of eggs from layers supplemented with feldspar show no significant differences from the control eggs, which supports the hypothesis that laying hens diets supplemented with low amounts (0.5% to 3%) of solid additives (i.e., biochar, feldspar) have no influence whatsoever on sensory traits of the eggs.

## 5. Conclusions

This study assessed supplementation of laying hens diet with BC and BC–aluminosilicates–glycerin mixture to lower the environmental impact while maintaining egg quality. The tested treatments had no statistically significant influence on $NH_3$ and VOC emissions, although the excreta N content was numerically lower. The dietary treatments had a positive effect on laying hens performance, egg quality and their sensory parameters. The results are consistent with other experiments in a similar field. However, there are very few investigations on feeding laying hens with biochar and effects of such dietary manipulation on birds' performance and excreta properties, including $NH_3$ and VOCs emissions, to compare and discuss.

More studies are still needed on BC feed supplementation. The BC itself is a wide "umbrella" term for material with a wide range of physicochemical properties. Research reporting BC feed additives should comprehensively report the properties as potentially having a significant impact on the key outcomes for the industry, environment and consumers.

**Author Contributions:** Conceptualization, S.O. and K.K.; formal analysis, D.K.; funding acquisition, K.K; investigation K.K. and D.K.; methodology, S.O. and M.K.; supervision, S.O. and M.K.; visualization K.K.; writing—original draft, K.K.; writing—review and editing, K.K., S.O. and J.A.K. All authors have read and agreed to the published version of the manuscript.

**Funding:** The work is financed by the Wrocław University of Environmental and Life Sciences under the research project number D220/0003/18 (Biochar as a Factor Mitigating the Emission of Odorous Substances from Poultry Production), as part of the "Innovative Doctorate" program. This research was partially supported by the Iowa Agriculture and Home Economics Experiment Station, Ames, Iowa. Project no. IOW05556 (Future Challenges in Animal Production Systems: Seeking Solutions through Focused Facilitation) sponsored by Hatch Act and State of Iowa funds).

**Acknowledgments:** We would like to thank Marek Kułażyński (Ekomotor, Ltd.) for providing the feed additives for the investigation.

**Conflicts of Interest:** The authors declare no conflict of interest. The funding agency had no influence on the experimental design, data analysis and conclusions.

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
