# Peer review of "Laying Hens Biochar Diet Supplementation—Effect on Performance, Excreta N Content, NH3 and VOCs Emissions, Egg Traits and Egg Consumers Acceptance"

_agriculture, doi:10.3390/agriculture10060237_

Round 1

Reviewer 1 Report

The authors improve the manuscript according the reviewer's suggestion

Author Response

Dear reviewer,

The text has been revised and the following changes have been implemented:

  • The keywords have been altered.
  • Some sentences have been reworded.
  • Information on the number of investigated eggs and age of the hens at the end of the experiment have been added.
  • Tables numbering has been revised.

Reviewer 2 Report

Dear authors,

thank you for your interesting study.

I have several comments for your manuscript.

Title:

I dont sure that abbrevations in title si correct (VOC´s). It would be possible to use whole term instead of an abbreviation?

Key words:

Don´t use the same words in key words and in title (biochar, laying hens, eggs). it's useless. Usually, the title and keywords are indexed.

Introduction

L66: ..could effect in an.. delete in

L67: broiler chickens instead chickens broiler

MaM

How long did the experiment take? From 20 weeks of age to how many weeks of age? Add this to manuscript.

I miss reference to Table 3 in 2.2. section

How many eggs were used for egg qwuality analysis? Please add this.

Table 2: Don´t use kcal. Please use kJ. kcal is not SI unit. Recalculate it.

It was the fiber content of the diet determined after the addition of biochar, which contains 75% fiber?

L199: Table 4 there should be. Not Table 3.

L212: 0.05.3?

L213: Results?

There are mistakes in L212-213. Please correct the text.

Results

L217: There should be Table 5

L243: There should be Table 6

L263: There should be Table 7

L291 and L306: why you write SD? Where is the SD values in table? Correct it, please.

L292: There should be Table 8

L307: There should be Table 9.

Conclusion

Please use short sentences in English. Not long sentences. The sentence (L397-399) is difficult to understand.

Good luck.

Sincerely,

reviewer.

Author Response

Dear Reviewer, thank you for your kind words. We have addressed all of your comments. Please refer to our responses below.

Title:

We have decided to replace “VOCs” with “odors” as amending VOCs to "Volatile Organic Compounds" would make the already very long title even longer. The "Volatile Organic Compounds" have been mentioned in the keywords in exchange.

Key words:

Thank you for this invaluable advice. The “biochar”, “laying hens” and “eggs“ keywords have been removed.

Introduction

L66: The sentence has been reworded.

L67: “Broiler chickens” are now in the sentence.

MaM

“For 13 weeks” has been added to the lines 98-99.

The reference to Table 3 has been added to the line 110.

“(86 eggs on average for the whole experiment)” has been added to the line 118.

“(A total of 375 eggs, 5 eggs per panelist with 1 egg per group)” has been added to the lines 120-121.

For other parameters, the information is already in the text (line 127).

Table 2: The value has been recalculated to 11297 kJ

It was the fiber content of the diet determined after the addition of biochar, which contains 75% fiber?

The fiber content described in the table is not the fiber after the addition of biochar. It this the fiber content of the basal (control) diet (line 135). The biochar was added to this diet in the amount of 1% and 2% by mass.

L199: L217: L243: L263: L292: L307: Thank you for the insight. The tables numbering has been corrected.

L212: L213: The artifacts have been removed.

Results

L291 and L306: The “SDs” have been removed from the text.

Conclusion

L397-399: The sentence was incorrectly structured and thus not understandable. It has been reworked.

Thank you!

This manuscript is a resubmission of an earlier submission. The following is a list of the peer review reports and author responses from that submission.

Round 1

Reviewer 1 Report

Comments to the Author

1 The Abstract section needs to be rewritten, and the purpose, treatment and results should be clear.

2 For laying hen experiments, there are too few experimental animals, hence the experimental data lack credibility.

3 The table format is not standardized, and the description of results is too redundant.

4 What is the purpose of the research?

Reviewer 2 Report

It is regrettable that the authors ability to produce text in English is low – a variety of semantic errors and misspellings, using ampersand instead of the word ‘and’, and syntactic constructions that violate the basic rules of constructing the sentences in English can be found in this paper, for instance: “…to let the treatment feeds to take effect.” [lines 113-114]; “Those 3-week periods took place three times over the course of the experiment.” [line 149]; “Manure samples were taken directly into the laboratory after collecting it and were put into…” [lines 167-168]; “Panelists’ mean grades for eggs’ appearance,…” [line 262]; “…shell weights were almost indifferent between the treatment groups and control.” [lines 334-335]. These are just a few examples. Moreover, expressions like “investigated laying hens”, “investigated additives”, “investigated treatments”, “laying hens’ manure/manures” (instead of droppings or caged laying hen excreta (manure free of litter)) or “amended feed/BC amendment” (instead of supplemented diets/BC supplementation) sound extremely unprofessional, to put it mildly. From my point of view, the authors also overuse the word research. Instead of ‘research’, less inflated synonyms are at our disposal e.g. study or experiment. But linguistic errors are not the only flaw in the submitted manuscript. 

After having carefully read the content, I find this article (manuscript ID: animals-792944) not suitable for publication within a section ‘Animal Nutrition’ of Animals run by the MDPI. There are more than a dozen reasons that lie behind such an opinion. Here I will focus on the main points.

  1. It is unclear from reading the Abstract (incoherent mishmash, in my opinion) whether the submitted manuscript deals more with the effects of dietary supplementation with biochar on egg performance in laying hens, eggshell quality and sensory attributes of hard-boiled eggs, or is more concerned with analysing the impact of BC dietary additions on excreta nitrogen content, and ammonia and VOC emissions. The aim of this study/article is unimaginative, and in fact incomprehensible. Instead of creating a clear purpose statement, the authors describe in details what has been done in their experiment.
  2. Reading title and the Introduction section, one might well conclude that the authors focus exclusively on biochar as dietary supplement, but in point of fact, two types of additives have been evaluated in their study: biochar alone (BC) and combined with clinoptilolite – a natural zeolite (BCM; 6 g clinoptilolite per kg feed in BCM2 dietary treatment). The positive effects of feeding clinoptilolite on the performance of laying hens and eggshell thickness are quite well recognised, but not a single word in this regard has been mentioned in the body of this manuscript, neither in the Introduction nor in the Discussion section. Moreover, there are no sensible explanations for choosing the applied dietary levels of BC and BCM.
  3. No information is provided (Animal use protocol) if the experiment was carried out with due regard to legislation governing the ethical treatment of animals. What is worse, due to the chaotic and insufficient description of materials and methods the manuscript seems to present unreliable data. Among other things, duration of the experimental phase (excluding four-week adaptation period) is not clearly defined, and nutrient content (metabolisable energy, crude protein, indispensable amino acids, Ca, P available) of the basal diet is not specified. Many important methodological details are either omitted (e.g. area of the eggshell where ‘crush test’ was applied) or presented in such sparse form (e.g. instead of brief description, the reference to the “Polish Standard Method”, unavailable for an international audience, is only made) that it is impossible to replicate the experiment – according to the principle that results have to be independently verifiable.
  4. The authors claim that based on peak heights “A total of 9 VOCs were identified in the analyzed gas samples…” including octan-3-one (an insect attractant and fragrance ingredient). It is puzzling, however, that none of the commonly targeted odorants, malodorous VFAs, DMA or TMA for example, was considered in this study.
  5. The results are presented in an unacceptably subjective manner. The statements of differences between treatment means should be backed up by statistical significance. Contrary to this rule, the authors emphasise numerical (percentage) differences in favour of the dietary supplements tested, as in the case of eggshell breaking strength or reduction of ammonia and VOC emissions (with ridiculous extenuation of the lack of significant differences for all TICs: "...most likely due to the high standard deviations [!!!] of the results.”). Besides, because of mixing up of the significance borders (p values <0.05 and <0.01 were considered simultaneously, as I guess, to make the manuscript ‘more scientific’) the information in tables is almost unreadable, all the more that not all treatment means within particular parameter are denoted by superscripts.
  6. The authors do not present concrete conclusions from their ‘study’, instead they complain that “there are very few investigations on feeding laying hens with biochar […] to compare and discuss with.” (pretty funny).

Overall, then, the authors created this paper to promote themselves rather than for the readers. More than 50% of their citations (14 out of 27 references) come from their own papers, which are, in most cases, not relevant to the subject of the manuscript.

Reviewer 3 Report

The manuscript in interesting and relevant as it deals with mitigating pollution in poultry farming and testing pollution reduction measures.

However, the paper needs revision on the following items:

Lines 34-35 „Investigated feed additives showed a positive effect on ammonia reduction (66 & 18%, after 1 h & 24 h, respectively); however, it was not statistically significant (p > 0.05).“ This statement is in conflict with the actual data presented in Table 4.

Lines 41-42 „Manure total nitrogen content was reduced due to the treatments (4 to 20%, p < 0.01)...“ Lower nitrogen content was not found in all the treated groups in comparison with control (Table 8).

Materials and Methods

The experimental design is not clearly written and need to be reworded by the authors, especially “Manure sampling”. How long did the trials last? Was there only one experiment or were there several different? What was the sequence of the experiments ? What investigations were carried out within separate experiments? There is no a clear enough picture of how the samples from control and treated groups were made up.

Results

In section Results there are included many data, which are available in the tables too. 

Footnotes to tables should be revised (Table 4. RVV-?, BC, „biochar group; BCM, biochar–aluminosilicates–glycerin mixture additive“; Table 6. RIV-?, “BC1 = 1% by mass of beech wood biochar; BC2 = 2% by mass of beech wood biochar; BCM1 = 1,5% by mass of beech wood biochar-based additive; BCM2 = 2% by mass of beech wood biochar-based additive“)

Table 4 indicates four control groups. Why? Why are the NH3 oncentration differences in the control groups so high?

Table 8. Under the influence of BC1 ammoniacal N (N-NH4+) increased, but under the influence of BC2 it decreased. Why ?

Conclusions

The conclusions are very general, not sufficiently informative and are inconsistent with the research results.

Line 360 „The use of biochar as a feed additive for laying hens has, in general, a positive effect“. These statement is true only for laying hens performance, egg quality and sensory parameters. The study did not indicate statistically significant influence of experimental diet on ammoniacal N, VOC, manure total N in BC2 and BCM1groups and differences in NH3.

Reviewer 4 Report

Line 48: do you have updated data to 2019?

Lines 65-73: provide relevant references

Use a different order to present M&M: 2.1 Laying hens and feed additives 2.2 Egg quality

Discussion: please avoid redundant phrase as “this is the first investigation on…” make the effort to discuss your results by comparing it with similar experiments in which other additives with similar mechanism have been used.

L310-315: can you compare data with other experimental design in poultry to improve the discussion of this section? Even if I agree with you that “to date, no recent publications have been found concerning the effect of dietary BC amendment on VOC emissions from laying hens or even poultry manure.” Probably there are other studies that evaluate the same parameters

L 339 “numerically higher” add numerically/numerical when not statistically significant

Lines 351-353: delete, you should improve your discussion by comparing your data with similar experiment (even with different additives but with similar action) in laying hen

Conclusions: should be more focused on your results